# 'Charge Reverse' Halogen Bonding Contacts in Metal-Organic Multi-Component Compounds: Antiproliferative Evaluation and Theoretical Studies

Subham Banik [1], Trishnajyoti Baishya [1], Rosa M. Gomila [2], Antonio Frontera [2,*], Miquel Barcelo-Oliver [2], Akalesh K. Verma [3,*], Jumi Das [3] and Manjit K. Bhattacharyya [1,*]

[1] Department of Chemistry, Cotton University, Guwahati 781001, India; chm2391001_subham@cottonuniversity.ac.in (S.B.); baishyatrishnajyoti@gmail.com (T.B.)
[2] Departament de Química, Universitat de les Illes Balears, Crta de Valldemossa km 7.7, 07122 Palma de Mallorca, Baleares, Spain; rosa.gomila@uib.es (R.M.G.); miquel.barcelo@uib.es (M.B.-O.)
[3] Cell & Biochemical Technology Laboratory, Department of Zoology, Cotton University, Guwahati 781001, India; jumidas979@gmail.com
* Correspondence: toni.frontera@uib.es (A.F.); akhilesh@cottonuniversity.ac.in (A.K.V.); manjit.bhattacharyya@cottonuniversity.ac.in (M.K.B.)

**Abstract:** Two new metal–organic multi-component compounds of Ni(II) and Co(II), viz. [Ni(3-CNpy)$_2$(H$_2$O)$_4$]ADS·2.75H$_2$O (**1**) and [Co(3-CNpy)$_2$(H$_2$O)$_4$](4-ClbzSO$_3$)$_2$ (**2**) (3-CNpy = 3-cyanopyridine, ADS = anthraquinone-1,5-disulfonate, 4-ClbzSO$_3$ = 4-chlorobenzenesulfonate), were synthesized and characterized using single crystal XRD, TGA, spectroscopic (IR, electronic) and elemental analyses. Both the compounds crystallize as multi-component compounds of Ni(II) and Co(II), with uncoordinated *ADS* and *4-ClbzSO$_3$* moieties in the crystal lattice, respectively. Crystal structure analyses revealed the presence of antiparallel nitrile···nitrile and π-stacked assemblies involving alternate coordinated *3-CNpy* and uncoordinated *ADS* and *4-ClbzSO$_3$* moieties. Moreover, unconventional charge reverse Cl···N halogen bonding contacts observed in compound **2** provide additional reinforcement to the crystal structure. Theoretical calculations confirm that the H-bonding interactions, along with anion–π(arene) and anion–π(CN) in **1** and π–π, antiparallel CN···CN and charge reverse Cl···N halogen bonds in **2**, play crucial roles in the solid state stability of the compounds. In vitro anticancer activities observed through the trypan blue cell cytotoxicity assay reveal that the compounds induce significant concentration dependent cytotoxicity in Dalton's lymphoma (DL) cancer cells, with nominal effects in normal healthy cells. Molecular docking studies reveal that the compounds can effectively bind with the active sites of anti-apoptotic proteins, which are actively involved in cancer progression.

**Keywords:** metal–organic multi-component compounds; charge reverse halogen bonding; combined NCI/QTAIM; trypan blue assay

## 1. Introduction

Research interest has surged in the investigation of transition metal coordination compounds that incorporate organic ligands, driven by their captivating structural arrangements and intricate network architectures [1–7]. The design, synthesis and development of metal–organic compounds with specific dimensionalities hinge upon diverse synthetic parameters, including metal–ion coordination with organic ligands, choice of solvent and reaction conditions [8–16]. Multi-component compounds, which are crystalline materials composed of two or more components (ions, atoms or molecules) in the same crystal lattice, have received remarkable attention in crystal engineering due to the evolving role of solid state chemistry in drug industries, electronic devices and synthetic organic

chemistry [17–21]. It has been firmly established that organic and metal–organic multi-component compounds offer novel solid formulations for active pharmaceutical ingredients, showcasing enhanced features like improved dissolution rates, thermal endurance or mechanical features [22–24].

Anthraquinone and its substituted derivatives have demonstrated efficacy in crafting coordination compounds, due to their ability to yield compounds with fascinating structural motifs with high thermal adherence topologies [25–27]. Anthraquinone disulfonate moieties possess the capability to serve as bridging ligands, owing to the existence of their two sulfonate groups, thereby offering versatility in coordination geometries [28–31]. In recent years, several anthraquinone-based compounds have been developed, and their significant anticancer properties were explored in different cancer cells [32–34]. Similarly, coordination compounds of 4-chlorobenzene and substituted derivatives have also attracted immense interest in research fields due to their various potential applications [35,36]. However, metal–organic compounds, wherein sulfonate derivatives act as counter anionic moieties within their crystal structures, have seldom been documented in the literature [37]. Metal–organic transition metal complexes involving cyanopyridines and nitrogen donor organic ligands also hold a unique place from a crystal engineering viewpoint, as they can provide interesting structure-guiding synthons to develop crystal structures with desired dimensionalities [38,39].

Non-covalent interactions are pivotal and constructive in metal–organic chemistry, crystal engineering and molecular biology, as well as in other domains of synthetic chemistry [40,41]. Numerous endeavors have been devoted to visualizing and quantifying the non-covalent synthons observed within supramolecular architectures due to their capacity to guide structures [42,43]. Non-covalent interactions, although inherently weaker than coordinate covalent bonds, exert significant influence on the construction of coordination network architecture, owing to their multitude and directional characteristics [44]. The exploration of the cooperative interplay of intermolecular hydrogen bonding and aromatic π-stacking interactions has become a compelling subject of inquiry [45,46]; as such, cooperative interactions are remarkably important for the stability of bio-molecules [47]. Interactions involving the nitrile functional groups of organic ligands are pivotal for the stabilization of supramolecular assemblies, as highlighted in prior research [48,49]. Just as observed with the carbonyl functional group in metal–organic compounds, the nitrile group has the potential to exhibit local dipole moments [50]; thus, organic functional groups engaged in dipole–dipole interactions may also serve as hydrogen bond acceptors [51]. Furthermore, the exploration of halogen bonding contacts in metal–organic compounds, which has been less frequently acknowledged, reveals new avenues in crystal engineering [52–55]. Halogen bonding interactions, a subset of σ-hole interaction, possess equally important roles in the stabilization of supramolecular assemblies [56–58]. However, reports of "charge reverse" halogen bonding, where the electron acceptor halogen atom is located on the electron-rich fragment (anion) and the electron donor atom is on the electron-poor fragment (cation), are still scarce in the literature.

Lattice anions serve not only as counter anionic motifs in coordination compounds, but they also engage in diverse novel interactions. These interactions, in turn, offer intricate dynamics in self-assembly processes [59,60]. The intricate interplay of supramolecular interactions between the counter anions and the H-bonded water cores is paramount. This dynamic interaction is instrumental in driving the spontaneous assembly of clusters that consist of anions and water molecules [61–64]. While there has been ample research on biologically relevant compounds featuring fascinating anion–water clusters [65–68], the investigation into supramolecular interaction involving sulfonate moieties, which are also biologically significant, has been overlooked.

In this research, we state the synthesis and crystal structures of two new multi-component metal–organic compounds incorporating Ni(II) and Co(II), viz. [Ni(3-CNpy)$_2$(H$_2$O)$_4$]ADS·2.75H$_2$O (**1**) and [Co(3-CNpy)$_2$(H$_2$O)$_4$](4-ClbzSO$_3$)$_2$ (**2**). In addition to elucidating the crystal structures, we characterized the compounds employing techniques

such as TGA, FT-IR and electronic spectroscopy, alongside elemental analysis. The crystal structures are reinforced by antiparallel nitrile···nitrile and π-stacked assemblies, featuring alternating coordinated *3-CNpy* and uncoordinated *ADS* alongside *4-ClbzSO₃* moieties. Notably, lattice water molecules and *ADS* anions within the crystal structure of compound **1** form interconnected layered anion–water clusters through non-covalent interactions, adding to their structural intricacy. In compound **2**, charge reversed Cl···N halogen bonding interactions are noticed, contributing to the structural rigidity in the layered assembly of the compound. We conducted theoretical investigations to examine the unconventional non-covalent interactions found in the compounds, focusing particularly on halogen bonds. The energetic attributes of the unusual supramolecular assemblies were theoretically examined, employing MEP surface analysis and the computational tools of combined QTAIM/NCI plotting. The compounds were subjected to in vitro anticancer assessment evaluation in the Dalton's lymphoma (DL) malignant cancer cell line, utilizing the trypan blue assay. To establish a correlation with the outcomes from the trypan blue assay, molecular docking was conducted to investigate the potential interactions of the compounds with anti-apoptotic BCL family proteins.

## 2. Results and Discussion

### 2.1. Synthesis and General Aspects

Compound **1** was isolated by reacting one equivalent of $Ni(CH_3COO)_2 \cdot 4H_2O$, two equivalents of 3-CNpy and one equivalent of $Na_2$-ADS in water at room temperature. In a similar way, compound **2** was obtained by the reaction between one equivalent of $Co(CH_3COO)_2 \cdot 4H_2O$, two equivalents of 3-CNpy and two equivalents of Na-4-ClbzSO₃ at room temperature in water. Compounds **1** and **2** are fairly soluble in water, and common organic solvents. Compounds **1** and **2** show room temperature (300 K) $\mu_{eff}$ values of 2.87 and 3.84 BM, respectively, suggesting the presence of two and three unpaired electrons per Ni(II) and Co(II) centers in the compounds, respectively [69].

### 2.2. Crystal Structure Analysis

The molecular structure of compound **1** is depicted in Figure 1. The bond lengths and bond angles around the Ni(II) center are summarized in Table S1. Compound **1** crystallizes in a monoclinic crystal system with space group $P2_1/c$. Crystal structure analysis reveals that the Ni(II) center in compound **1** is six coordinated with two mono-dentate 3-CNpy and four coordinated water molecules.

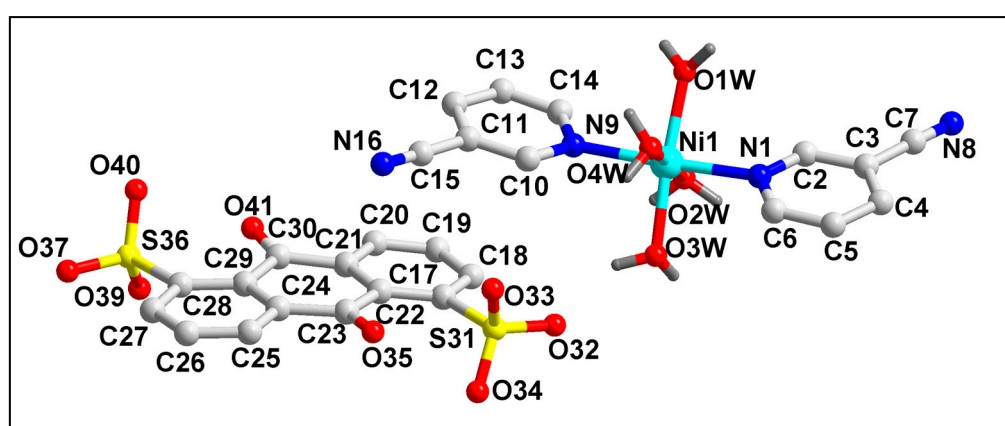

**Figure 1.** Molecular structure of [Ni(3-CNpy)₂(H₂O)₄]ADS·2.75H₂O (**1**). The lattice water molecules are omitted for clarity.

The presence of one uncoordinated divalent ADS moiety compensates the di-positive charge of the overall complex moiety. Moreover, three lattice water molecules are also present in the asymmetric unit of the compound. The coordination geometry around the Ni(II) center is a distorted octahedron, with the equatorial positions occupied by

four coordinated water molecules (O1W-O4W); in contrast, theN1 and N9 atoms from the coordinated 3-CNpy moieties occupy the axial sites. The four equatorial atoms, viz. O1W, O2W, O3W and O4W, are distorted from the mean equatorial plane with a mean r.m.s. deviation of 0.0311 Å. The average Ni–O and Ni–N bond lengths were found to be similar with the previously reported Ni(II) compounds [70].

An analysis of the crystal structure of compound **1** unfolds the formation of 1D supramolecular chain along the crystallographic a axis, aided by anti-parallel CN⋯CN π-stacked and C–H⋯N hydrogen bonding interactions (Figure S1). The nitrile moieties of adjacent monomeric units are involved in anti-parallel CN⋯CN π-stacking interactions with C15⋯N8 and C7⋯N16 separation distances of 3.61 and 3.57 Å, respectively [71]. The corresponding C13−C15≡N16 and C3−C7≡N8 angles of the coordinated 3-CNpy moieties in compound **1** deviate slightly from linearity (178.4 and 178.2°), which may be an outcome of the non-covalent interactions of the nitrile moieties attached to the aromatic rings [72]. In addition, C–H⋯N hydrogen bonding interactions are also observed, which involve the nitrile groups of 3-CNpy having C4–H4⋯N16 and C12–H12⋯N8 distances of 2.52 and 2.63 Å, respectively (Table 1).

**Table 1.** Selected hydrogen bond distances (Å) and angles (°) for compounds **1** and **2**.

| D–H⋯A | d(D–H) | d(D⋯A) | d(H⋯A) | <(DHA) |
|---|---|---|---|---|
| **1** | | | | |
| C5–H5⋯O35#1 | 0.95 | 2.37 | 3.141(3) | 137.9 |
| C13–H13⋯O40#2 | 0.95 | 2.45 | 3.144(3) | 130.0 |
| O1W–H1WA⋯O5W#3 | 0.87 | 1.81 | 2.592(4) | 148.9 |
| O1W–H1WB⋯O32#4 | 0.87 | 1.84 | 2.705(3) | 175.6 |
| O2W–H2WA⋯O8W | 0.87 | 1.90 | 2.769(11) | 174.5 |
| O2W–H2WA⋯O9W | 0.87 | 1.91 | 2.743(10) | 159.4 |
| O2W–H2WB⋯O40#5 | 0.87 | 1.83 | 2.700(3) | 174.4 |
| O3W–H3WA⋯O32 | 0.87 | 1.91 | 2.694(3) | 149.6 |
| O3W–H3WB⋯O37#5 | 0.87 | 1.93 | 2.799(3) | 171.8 |
| O4W–H4WA⋯O6W | 0.87 | 1.90 | 2.751(10) | 166.4 |
| O4W–H4WA⋯O7W | 0.87 | 1.90 | 2.733(11) | 160.8 |
| O4W–H4WB⋯O37#6 | 0.87 | 1.81 | 2.675(3) | 172.4 |
| C27–H27⋯O5W#7 | 0.95 | 2.40 | 3.185(5) | 140.0 |
| O6W–H6WA⋯O33 | 0.87 | 1.81 | 2.669(8) | 167.1 |
| O8W–H8WA⋯O34#8 | 0.87 | 2.00 | 2.857(12) | 170.0 |
| O8W–H8WB⋯O4W#9 | 0.87 | 2.21 | 3.073(12) | 173.8 |
| O9W–H9WA⋯O34# | 0.87 | 1.95 | 2.776(9) | 157.1 |
| O9W–H9WB⋯O4W#9 | 0.87 | 2.26 | 3.009(11) | 144.9 |
| O5W–H5WA⋯O6W | 0.87 | 1.59 | 2.41(2) | 155.5 |
| O5W–H5WB⋯O39#6 | 0.87 | 1.96 | 2.780(4) | 156.9 |
| O7W–H7WA⋯O7W#1 | 0.87 | 1.35 | 2.19(4) | 162.1 |
| O7WH7WB⋯O33 | 0.87 | 1.92 | 2.765(11) | 164.8 |
| **2** | | | | |
| C3A–H3A⋯N8 | 0.93 | 3.950 | 2.99 | 168.1 |
| C4–H4⋯Cl1A | 0.93 | 3.943 | 2.86 | 165.2 |
| C5–H5⋯N8 | 0.93 | 3.460 | 2.64 | 146.4 |
| C7A–H7A⋯O9A | 0.93 | 3.644 | 2.92 | 135.1 |
| O1W–H1WA⋯O10A | 0.87 | 2.757(2) | 1.90 | 170.6 |
| O1W–H1WB⋯O9A#10 | 0.87 | 2.770(2) | 1.90 | 176.0 |
| C2–H2⋯O9A | 0.93 | 3.531 | 2.69 | 147.2 |

#1 1 − X, 1 − Y, 2 − Z; #2 − X, 1 − Y, 1 − Z; #3 +X, 3/2 − Y, −1/2 + Z; #4 1 − X, 1/2 + Y, 3/2 − Z; #5 1 + X, +Y, +Z; #6 −X, 1/2 + Y, 3/2 − Z; #7 −X, 1 − Y, 2 − Z; #8 +X, 1/2 − Y, −1/2 + Z; #9 1 − X, −1/2 + Y, 3/2 − Z; #10 1 − X, 1 − Y, 2 − Z.

The coordinated 3-CNpy and uncoordinated ADS ligands of compound **1** are involved in the formation of the π-stacked ternary assembly (Figure 2a). The phenyl and pyridine rings of ADS and 3-CNpy are involved in $(\pi\text{-}\pi)_1$, $(\pi\text{-}\pi)_2$ and $(\pi\text{-}\pi)_3$ interactions having centroid(C17–C22)–centroid(N1, C2–C6), centroid(N1, C2–C6)–centroid(C26–C29, C23, C24) and centroid(C26–C29, C23, C24)–centroid(N9, C10–C14) separations of 3.96, 3.88 and 3.99 Å, respectively. The slipped angles (angle between the ring normal and the vector joining the ring centroids) were found to be 20.7, 20.7 and 20.6°, respectively [73]. Moreover, O–H···O hydrogen bonding interactions were also observed involving the coordinated water molecules (O1W–H1WA···O33 = 1.90 Å; O2W–H2WB···O33 = 1.83 Å; O3W–H3WA···O37 = 1.93 Å; O4W–H4WB···O40 = 2.40 Å). A closer look also reveals the presence of two different types of anion–π, viz. anion–π(arene) and anion–π(CN), interactions in the compound involving the π electrons of the aromatic ring and –CN fragments of the 3-CNpy, respectively (Figure 2b).

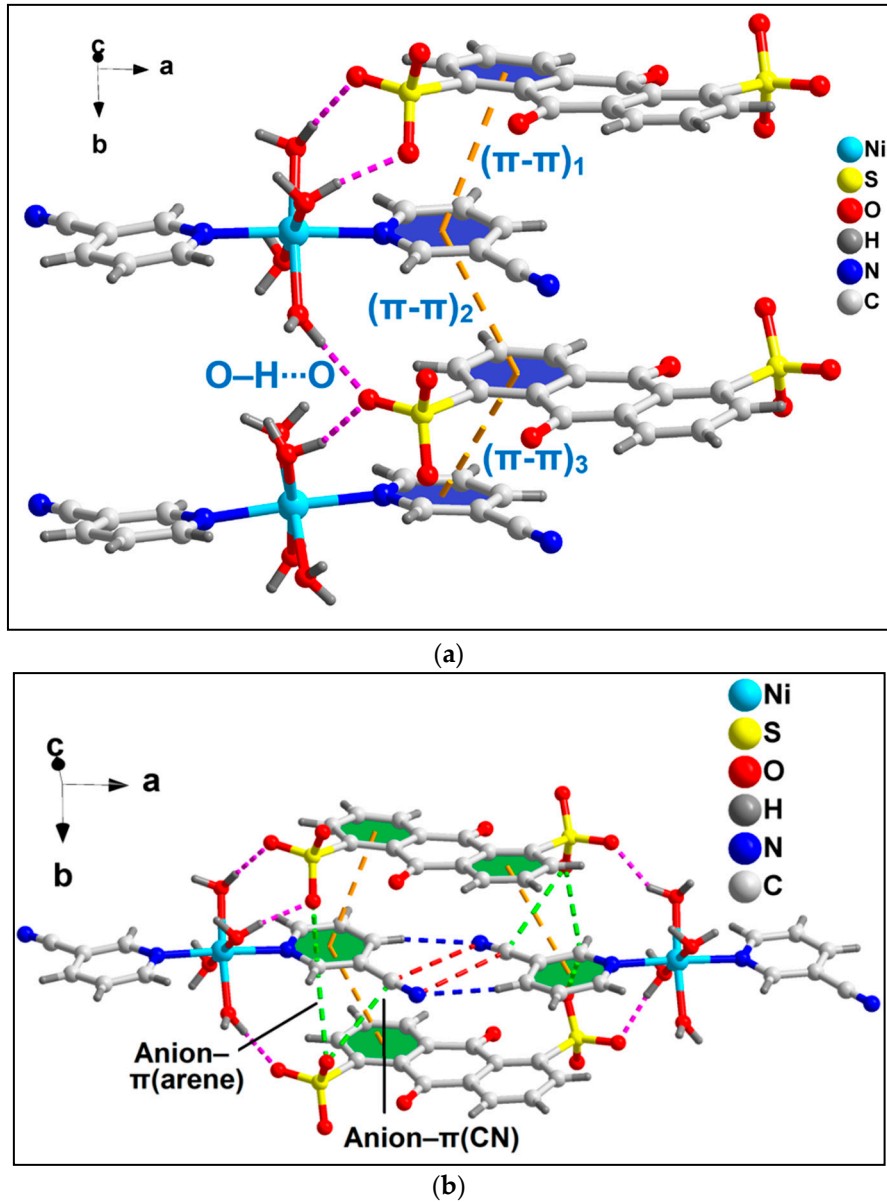

(**a**)

(**b**)

**Figure 2.** (**a**) π-stacked ternary assembly involving the coordinated 3-CNpy and uncoordinated ADS moieties; (**b**) two different types of anion–π interactions of compound **1**, viz. anion–π(arene) and anion–π(CN).

The anion–π(arene) interaction involves the π system of the pyridine ring of the coordinated 3-CNpy having O33···C14 and O37···C14 distances of 2.81 and 3.17 Å, respectively, while the anion–π(CN) interaction involves the π system of the nitrile group of the same 3-CNpy moiety having an O37···C15 distance of 3.31 Å. The aforementioned anti-parallel π-stacked CN···CN and C–H···N hydrogen bonding interactions also play important roles in the formation of the supramolecular assembly. These supramolecular ternary assemblies propagate to form a layered architecture along the crystallographic ab plane (Figure 3). These interactions have been further characterized theoretically via various computational tools (vide infra).

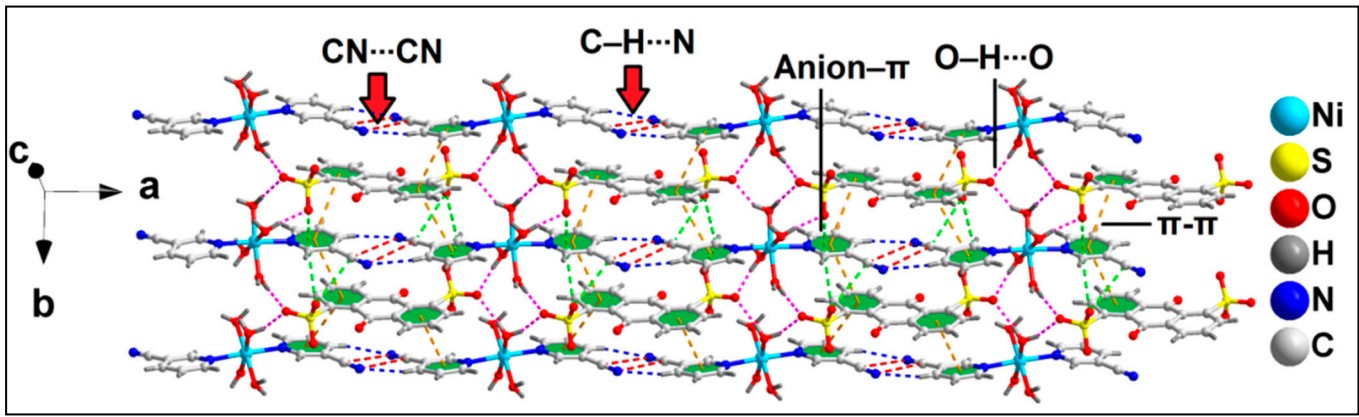

**Figure 3.** Layered assembly of compound **1** assisted by uncoordinated ADS anions along the ab plane.

The molecular structure of compound **2** is depicted in Figure 4. Selected bond lengths and bond angles around the Co(II) center are summarized in Table S1. Compound **2** crystallizes in a monoclinic crystal system with space group C2/m. In compound **2**, the Co(II) center is hexa-coordinated with two 3-CNpy ligands, along with four coordinated water molecules. The di-cationic charge of the cationic complex is neutralized by the two uncoordinated 4-ClbzSO₃ anions in the crystal lattice. The coordination geometry around the Co(II) center is an ideal octahedron where the nitrogen atoms (N1, N1′) of 3-CNpy occupy the axial sites, and the water molecules are present at the equatorial positions with zero r.m.s. deviation. Crystal structure analysis reveals the presence of a two-fold axis of symmetry in the crystal structure, with the axis passing through the Co(II) center. The Co–N and Co–O bond lengths are well consistent with the previously reported Co(II) compounds [74–77].

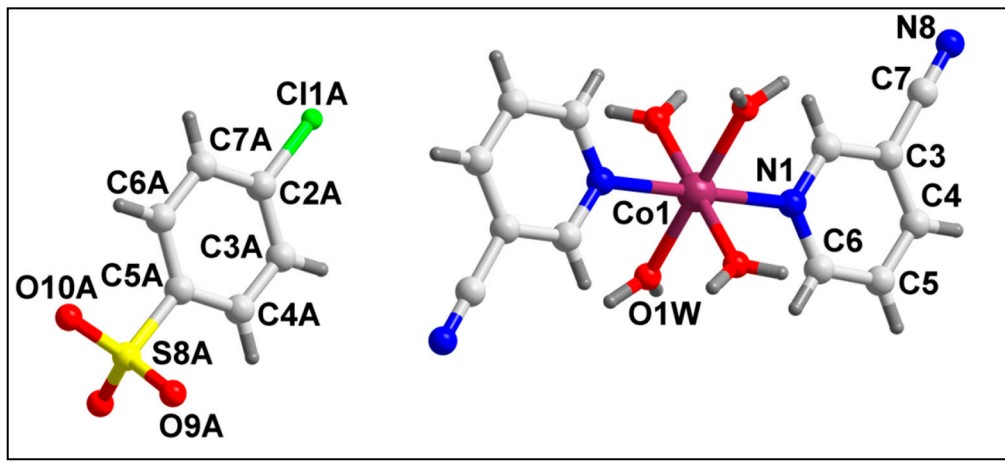

**Figure 4.** Molecular structure of [Co(3-CNpy)₂(H₂O)₄] (4-ClbzSO₃)₂ (**2**).

Crystal structure analysis of compound **2** reveals the formation of 1D supramolecular chain along the crystallographic b axis assisted by π-stacking and O–H···O and C–H···O hydrogen bonding interactions (Figure 5). Aromatic π-stacking interactions are observed between the aromatic rings of coordinated 3-CNpy and uncoordinated 4-ClbzSO$_3$ moieties having a centroid(N1, C2–C6)–centroid(C2A–C7A) distance of 3.97 Å, with the closest C–C distance of 3.47 Å (C3–C7A). The coordinated water molecules and O atoms of uncoordinated 4-ClbzSO$_3$ moieties are involved in O–H···O hydrogen bonding interactions, with O1W–H1WA···O10A and O1W–H1WB···O9A distances of 1.89 and 1.90 Å, respectively. C–H···O hydrogen bonding interactions with C2–H2···O9 distances of 2.69 Å are also observed.

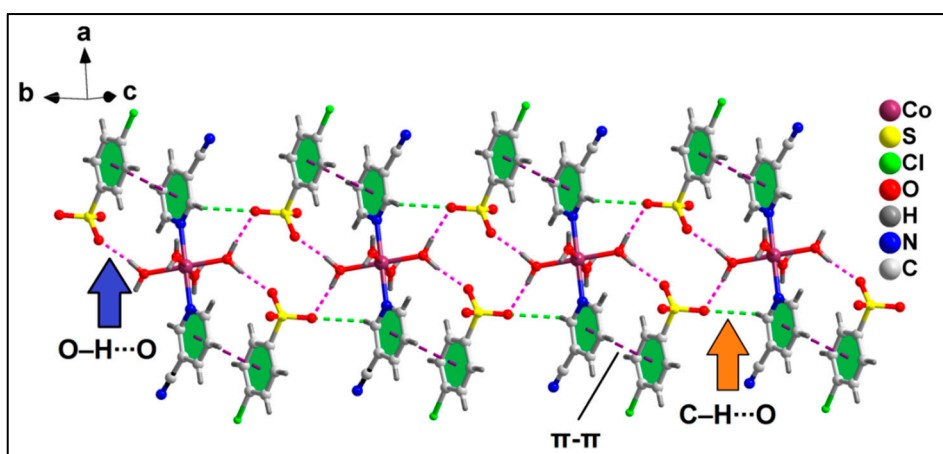

**Figure 5.** 1D supramolecular chain of compound **2** assisted by O–H···O, C–H···O and π–π stacking interactions.

The neighboring 1D chains of compound **2** are interconnected via C–H···N and C–H···Cl hydrogen bonding interactions to form the layered assembly along the crystallographic ab plane (Figure 6). C–H···N hydrogen bonding interactions are observed in the layered assembly involving the N atom of the –CN moiety of 3-CNpy, which has a C5–H5···N8 distance of 2.64 Å. Moreover, C–H···O hydrogen bonding interactions are also observed involving the O atom of 4-ClbzSO$_3$, with a C7A–H7A···O9A distance of 2.92 Å.

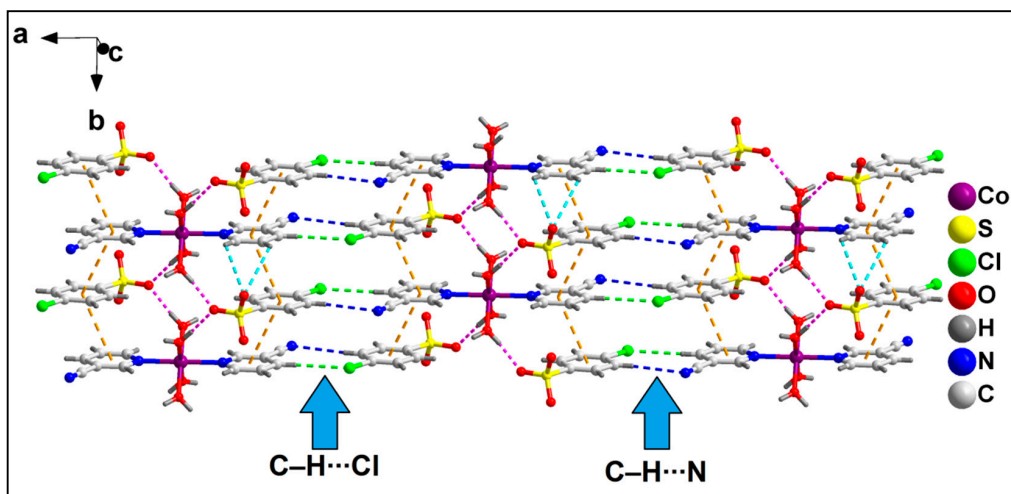

**Figure 6.** Layered assembly of compound **2** along the crystallographic ab plane.

Further analysis reveals the formation of the supramolecular hexamer of compound **2** that involves two Co(II) complex cationic moieties and four anionic 4-ClbzSO$_3$ moieties

(Figure 7). The hexameric assembly is formed by aromatic π-stacking, O–H···O hydrogen bonding and unconventional Cl···N halogen bonding interactions. The N8 atom of the cyano group of the 3-CNpy moiety is involved in unconventional Cl···N halogen bonding interactions [78] with the Cl1A atom of 4-ClbzSO$_3$, with a Cl1A···N8 separation distance of 3.39 Å. The corresponding C7–N8···Cl1A angle is 178.9°, indicating the strong nature of the interaction [79]. Such Cl···N halogen bonding interactions in coordination compounds have been scarcely found in the literature. Aromatic π-stacking interactions are observed between the aromatic rings of coordinated 3-CNpy and uncoordinated 4-ClbzSO$_3$ moieties, having a centroid(N1, C2–C6)–centroid(C2A–C7A) distance of 3.79 Å. O–H···O hydrogen bonding interactions are also observed involving the coordinated water molecule and O atoms of the uncoordinated 4-ClbzSO$_3$ moiety, with an O1W–H1WA···O10A distance of 1.89 Å. The energetic feature of this hexameric assembly of the compound has been further studied theoretically (vide infra).

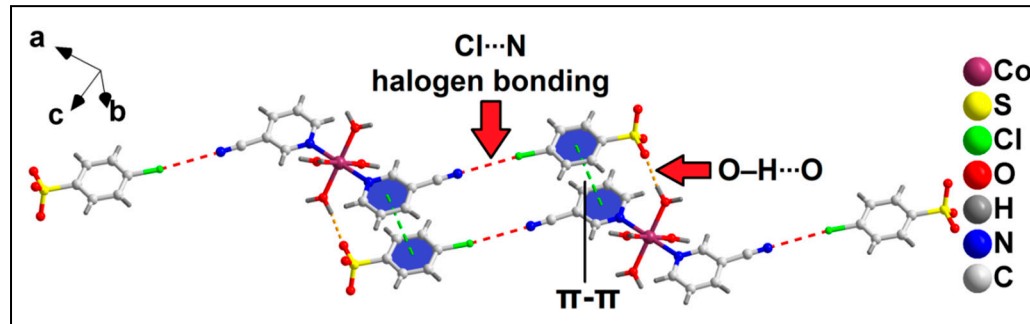

**Figure 7.** Formation of supramolecular hexameric assembly of compound **2** assisted by Cl···N halogen bonding, π-stacking interaction and O–H···O hydrogen bonding.

### 2.3. Spectral Studies

#### 2.3.1. FT-IR Spectroscopy

The FT-IR spectra of compounds **1** and **2** have been recorded in the region of 4000–500 cm$^{-1}$ (Figure S2). The broad absorption bands in the region of 3200–3510 cm$^{-1}$ can be attributed to the O–H stretching vibrations of the coordinated/lattice water molecules [80]. FT-IR spectra of the compounds also exhibit absorption bands due to $\rho_r$ (H$_2$O) (710 cm$^{-1}$) and $\rho_w$ (H$_2$O) (636 cm$^{-1}$), corroborating the presence of coordinated water molecules [81]. The absorption peaks at approximately 2240 cm$^{-1}$ in the spectra of both the compounds can be attributed to the ν(C-N) stretching vibrations of the 3-CNpy moieties [82]. The similar position of the peak with that of the free ligand suggests that the nitrile group of 3-CNpy is not coordinated to the respective metal centers [83]. The coordination of the 3-CNpy ligands with the metal centers through the pyridine ring N-atom can be corroborated with the shifting of ring stretching frequencies of the 3-CNpy [84]. The strong absorption bands at 1643 cm$^{-1}$ in the FT-IR spectrum of compound **1** are due to the presence of the ketonic group of ADS [85]. The stretching vibrations of the sulfonate groups of ADS and 4-ClbzSO$_3$ are obtained at 1270–1180 ($\nu_{as}$) and 1042 ($\nu_s$) cm$^{-1}$ in the FT-IR spectra of the compounds, respectively [86].

#### 2.3.2. Electronic Spectroscopy

The electronic spectra of the compounds were recorded in both the solid and aqueous phases (Figures S3 and S4). The spectra of the compounds reveal the presence of distorted octahedral Ni(II) and Co(II) centers in compounds **1** and **2**, respectively [87–91]. The absorption peaks for the π→π* transition of the aromatic ligands were obtained at their expected positions [92–94].

### 2.4. Thermogravimetric Analyses

The thermogravimetric curves of compounds **1** and **2** were recorded in the temperature range of 25–1000 °C under an $N_2$ atmosphere at a heating rate of 10 °C/min (Figure S5). In the case of compound **1**, the thermal decomposition of 17.49% in the temperature range of 50–115 °C (calcd. = 16.59%) can be attributed to the loss of lattice and coordinated water molecules [95,96]. In the temperature range of 116–533 °C, the uncoordinated ADS moiety decomposed with the observed weight loss of 48.07% (calcd. = 50.02%) [97]. In the temperature range of 534–695 °C, two coordinated 3-CNpy moieties decomposed (obs. = 24.21%; calcd. = 26.11%) [97]. For compound **2**, the water molecules that are coordinated decomposed in the 40–110 °C temperature range, with a weight loss of 13.28% (calcd. = 11.96%) [95,96]. In the temperature range of 111–350 °C, the 24.9% weight loss can be attributed to the loss of the 4-ClbzSO$_3$ moiety (calcd. = 26.52%) [98]. Finally, in the third step, two 3-CNpy and 4-ClbzSO$_3$ moieties decomposed with a weight loss of 55.91%, observed in the temperature range of 351–840 °C (calcd. = 55.31%) [97,98].

### 2.5. Theoretical Study

The theoretical study focused analysis on the hydrogen bond, halogen bond and anion–π and π-stacking interactions, which are important in understanding the crystal packing of compounds **1** and **2**. We are particularly intrigued by the Cl···N halogen bonds (HaB), since they are established between a pair of ions of opposite sign where the electron acceptor halogen atom is located on the electron-rich side (anion), and the electron donor (N-atom) is on the electron-poor side (cation). Therefore, these contacts can be termed as "charge reverse halogen bonds", the opposite term to the well-known charge-assisted hydrogen bonds [99].

In order to use neutral systems, we initially computed the MEP surfaces of the ion pairs of **1** and **2**. The MEP surfaces of compound **1**, where we have used two different orientations, are represented in Figure 8. The figure shows that the MEP maximum is located at the H-atoms of the Ni-coordinated water molecules (+111 kcal/mol), and the minimum is at the sulfonate groups (−59 kcal/mol). The MEP is also large and negative at the carbonyl group (−58 kcal/mol), likely due to the influence of the adjacent sulfonate group. The MEP values over the coordinated cyanopyridine rings are positive (+43 to +47 kcal/mol) and negative (−9 to −24 kcal/mol) over the anthraquinone system, thus explaining the strong tendency to form π-stacking assemblies. Finally, it is worth mentioning that the MEP value at the N-atom of the cyano group is negative (−13 kcal/mol), while it is positive at the adjacent H-atom (+49 kcal/mol), also supporting the formation of the C–H···N hydrogen bonds.

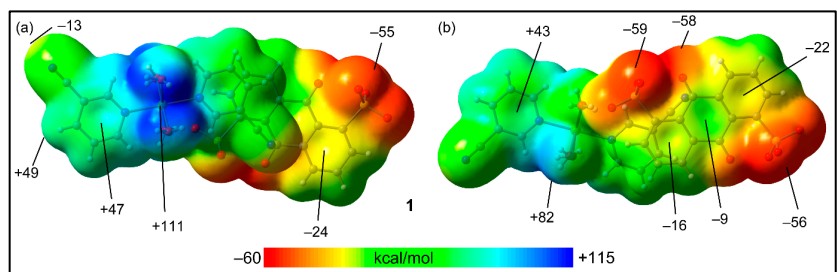

**Figure 8.** MEP surfaces of compound **1** (two different orientations), where the cationic unit is on top (**a**) and the anionic unit is on top (**b**) at the RI-BP86-D3/def2-TZVP level of theory. Isovalue 0.001 a.u. The energies at selected points are indicated in kcal/mol.

The MEP surfaces of compound **2** are shown in Figure 9 using two different trimeric assemblies (HaB, neutral systems). In the case of the planar assembly (Figure 9a), it can be observed that the MEP maximum is located at the H-atoms of the coordinated water molecules (+99 kcal/mol), and the minimum is located at the sulfonate groups (−53 kcal/mol), similar to compound **1**. The MEP values over the 3-CNpy ring are large and positive; over the 4-ClbzSO$_3$ ring, the value is negative (−19 kcal/mol).

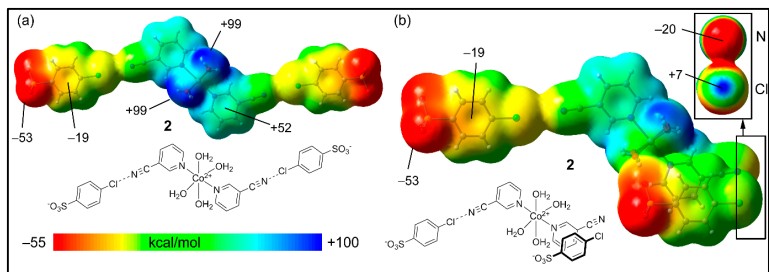

**Figure 9.** MEP surfaces of compound **2**, where the trimer is HaB (**a**) or HaB + π-stacked (**b**) at the RI-BP86-D3/def2-TZVP level of theory. Isovalue 0.001 a.u. The energies at selected points are indicated in kcal/mol.

In order to investigate the existence of an σ-hole at the Cl-atom, we used the neutral trimer (HaB + π-stacked) shown in Figure 9b, where the anisotropy at the Cl-atom can be studied. Interestingly, in spite of the global negative charge of the 4-ClbzSO₃, there is a modest σ-hole at the Cl-atom on the extension of the C–Cl bond (+7 kcal/mol), which can be appreciated in the amplification of the surface (top-right part of the figure) where the Cl and CN groups are stacked. For this representation, we used a different scale ($\leq$−20 to +7 kcal/mol). In addition, the N-atom of the cyano group is negative in spite of the dicationic nature of the Co(II) complex. This MEP analysis confirms that the "charge reverse" halogen bonds are electrostatically favored.

Figure 10 represents the QTAIM and NCI plot analyses combined to study the intricate combination of contacts that are formed in the double π-stacking assembly formed between two anionic and cationic moieties of compound **1**. The very large formation energy (−119.9 kcal/mol) obtained is due to the strong electrostatic effects. The combined QTAIM/NCI plot study confirms the presence of strong H-bonds between the coordinated water molecules and the sulfonate groups, which are characterized by bond critical points (CPs, red spheres), bond paths (orange lines) and blue RDG isosurfaces. The study also confirms the presence of anion–π(arene) and anion–π(CN) interactions characterized by bond CPs, bond paths and green RDG isosurfaces located between the anion and the C-atoms of the arene or the cyano groups. The antiparallel CN···CN interaction is also confirmed by a bond CP, bond path and green isosurface located between both groups. We evaluated the contribution of the H-bonds using the $V_r$ predictor and the equation proposed by Espinosa et al. [100], showing that the stabilization of the assembly due to these O–H···O H-bonds is −40.5 kcal/mol. This strong contribution confirms the relevance of these interactions in the X-ray packing. The C–H···N contacts are significantly weaker, with a total contribution of only −2.6 kcal/mol.

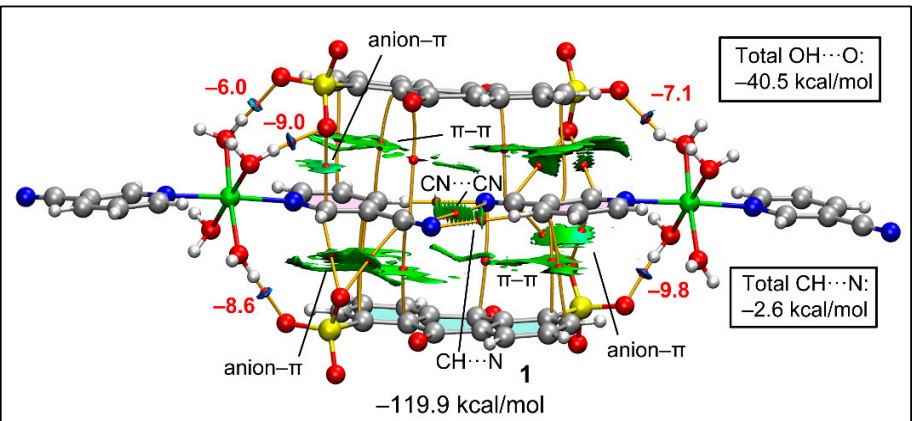

**Figure 10.** Combined QTAIM (bond CPs in red and bond path as orange lines) and RDG isosurface of the tetrameric assembly of compound **1**. The energies of the H-bonds are given in red adjacent to the bond CPs.

Figure 11 shows the six-component assembly of compound **2** used for the theoretical analysis that is formed by the interaction of two Co(II) complexes and four 4-ClbzSO$_3$ counter ions. We computed the formation energy of the hexameric assembly using two distinct trimers as the starting point, as represented in Figure 11. In the first case (Figure 11, left), the assembly of the halogen bonded trimer accounts for the formation of the H-bonds and π–π and Cl···CN interactions, all characterized by the corresponding bond CPs, bond paths and green/blue RDG isosurfaces.

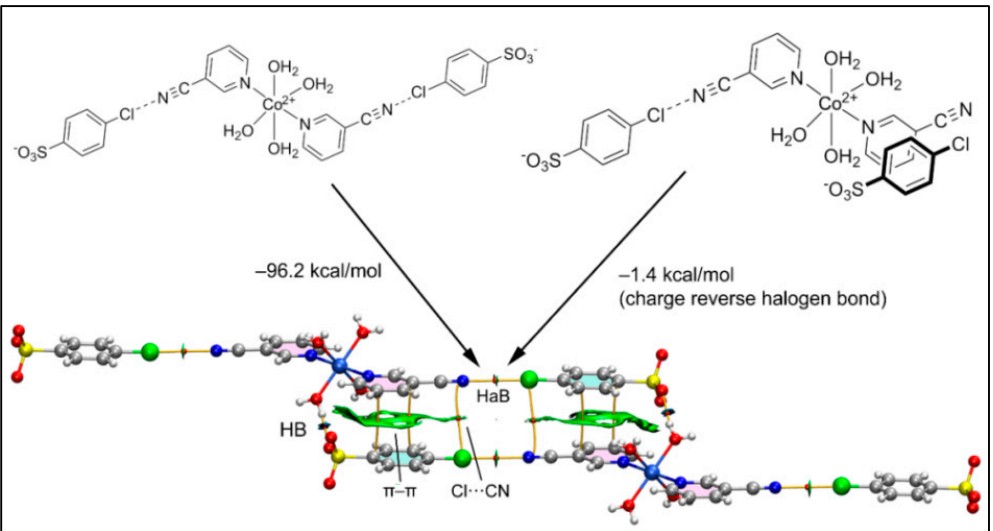

**Figure 11.** Combined QTAIM (bond CPs in red and bond path as orange lines) and RDG isosurface of the hexameric assembly of compound **2** and the formation energies computed from the self-dimerization of two different trimers.

The very large dimerization energy ($-96.2$ kcal/mol) obtained is basically due to the strong electrostatic attraction between the sulfonate and Co(II) complex, which are disposed in close proximity. In sharp contrast, the second assembly (Figure 11, right) exhibits a very modest dimerization energy ($-1.4$ kcal/mol) corresponding to the contribution of the charge reverse halogen bonds; these are revealed by the QTAIM analysis that indicates a bond CP and bond path connecting the Cl and N-atoms. This quite modest interaction energy ($-0.7$ kcal/mol for each Cl···N) of the HaB interaction agrees well with the small MEP value at the σ-hole of the Cl-atom and the small RDG green isosurface coincident to the locations of the bond CPs that characterize the HaBs.

### 2.6. Cytotoxicity Assay Using Trypan Blue Method

The trypan blue assay is based on the fact that living cells with intact cell membranes do not allow trypan blue dye to enter the cell membrane; in contrast, it can enter the damaged cell membrane of non-viable cells and stain as blue. As a result, when trypan blue dye is added to a treated cell culture, the non-viable cells appear as blue, and viable cells appear as clear/unstained [101]. In this study, the trypan blue assay revealed that the compounds are able to induce concentration dependent cytotoxicity in DL cells (Figures 12 and 13), with nominal effects in normal healthy PBMC cells. Comparative analysis reveals that compound **1** induced higher cytotoxicity than that of compound **2** after 24 h of treatment. The findings obtained for the compounds were analyzed comparatively with the reference drug cisplatin [102], which shows higher cytotoxicity than the compounds. The half maximum inhibitory concentration (IC$_{50}$), defined as the concentration required for 50% cell death in a cell culture, is often used to evaluate the cytotoxic potency of newly developed compounds/drugs. Figure 14 represents the dose response curves and the IC$_{50}$ values for the compounds and cisplatin. Compound **1** exhibits a lower IC$_{50}$ value (43 μM) as compared to compound **2** (78 μM) in the DL cell line after 24 h of treatment (Figure 14).

The IC$_{50}$ value of the reference drug, cisplatin, was found to be 0.63 µM in the same experimental setup.

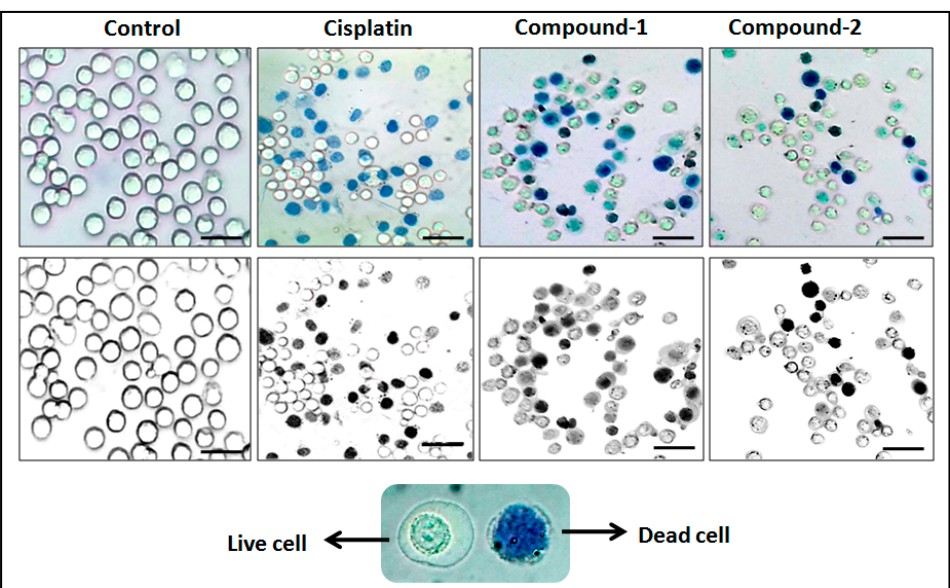

**Figure 12.** Compounds **1** and **2** mediated cytotoxicities in DL cell line after 24 h treatment at 10 µM concentration. Lower panel shows that the black and white images of the upper panel have clearer visualization between stained and unstained cells.

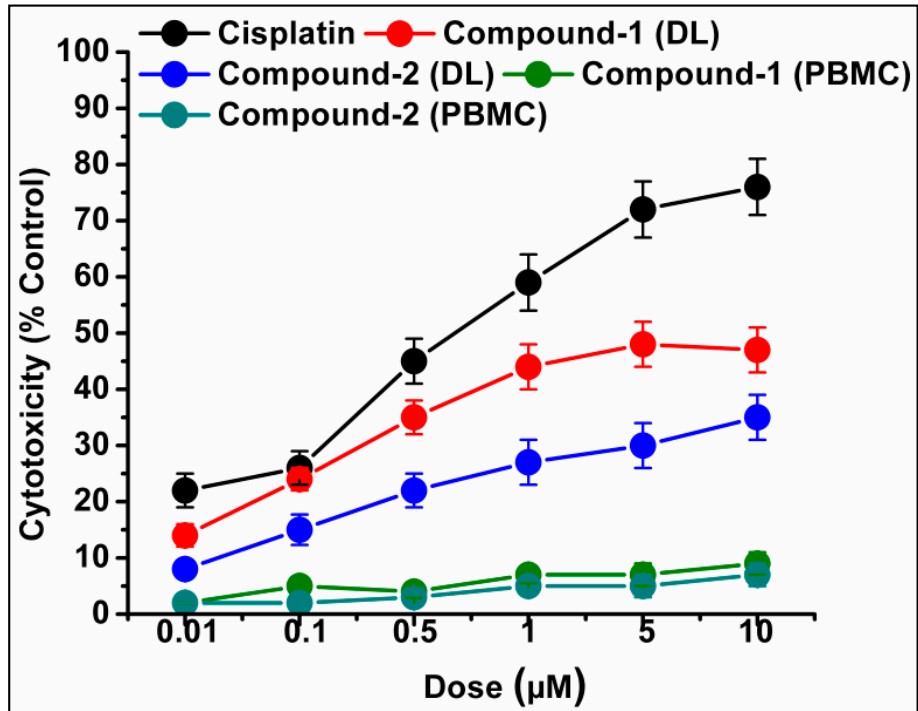

**Figure 13.** Cell cytotoxicity percentages induced by compounds **1** and **2** in DL cells after exposure with various test concentrations. Cisplatin was used as reference drug. Data are mean ± S.D., *n* = 3.

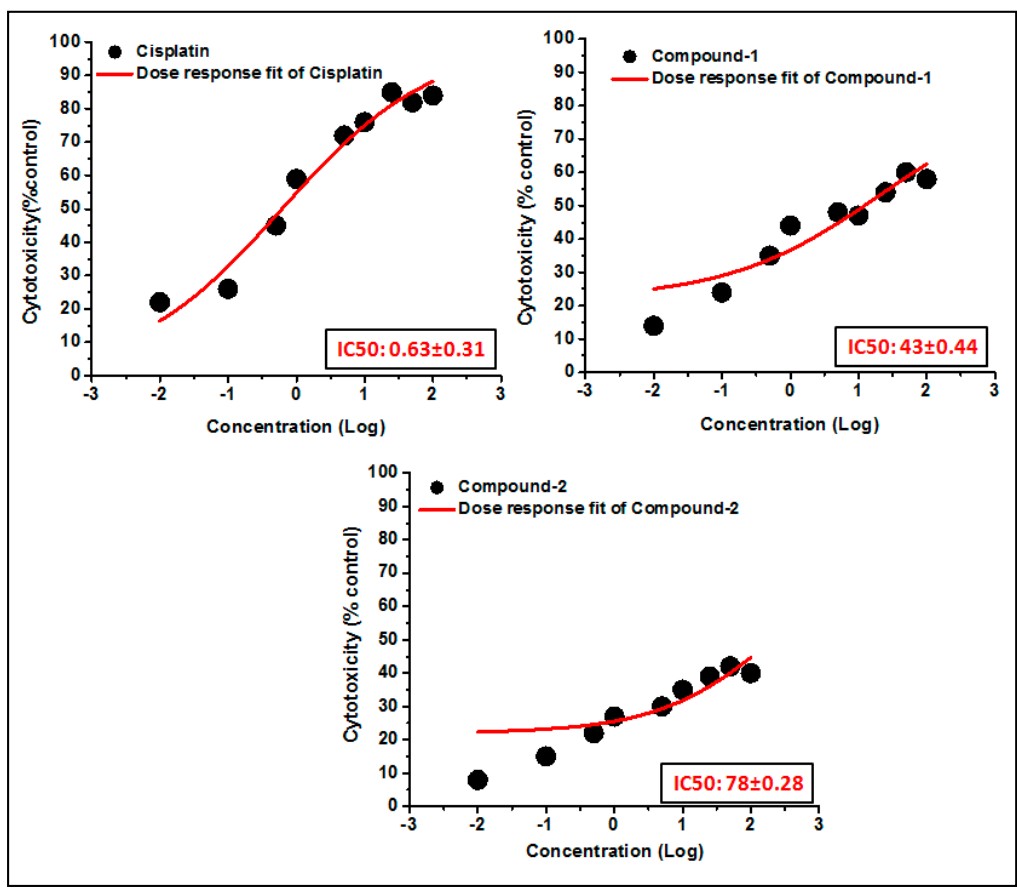

**Figure 14.** The dosage response curves of compounds **1** and **2** are shown in order to determine the IC$_{50}$ values in DL and PBMC cell lines.

*2.7. Molecular Docking Simulation*

Molecular docking is a very useful approach in the discovery of drugs; it is useful for identifying and optimizing drug molecules, predicting the binding affinity and selectivity of the molecules, and designing new drugs with improved pharmacological properties [103]. The findings obtained from the trypan blue assay encouraged us to carry out a molecular docking study that considered BCL family proteins, viz. BCL-2 and BCL-XL. BCL-2 and BCL-XL have been significantly reported to play an active role in various stages of cancer progression and metastasis [104]. From our study, it was revealed that both compounds **1** and **2** possess strong binding affinities for BCL-2 and BCL-XL target proteins (Figures 15 and 16). Compound **1** interacts with the BCL-2 receptor protein via four H-bonds, with active amino acid sites Tyr19, Lys20, Glu149 and Ser102. On the other hand, compound **2** also interacts with the BCL-2 receptor via four hydrogen bonds with Tyr19, Lys20 and Ser102. Moreover, compound **1** interacts with the BCL-XL receptor via seven hydrogen bonds with His177, Asn128, Glu124, Tyr173 and Trp169, whereas compound **2** interacts via five hydrogen bonds with Trp169, Tyr173, Glu124 and Asn128 residues. Therefore, the results suggest that both compounds have significant cytotoxicity inducing abilities due to their effective interactions with target receptors. Comparatively, compound **1** interacts more efficiently with the target proteins as compared to compound **2**, which may be responsible for its higher cytotoxicity (Figure 17).

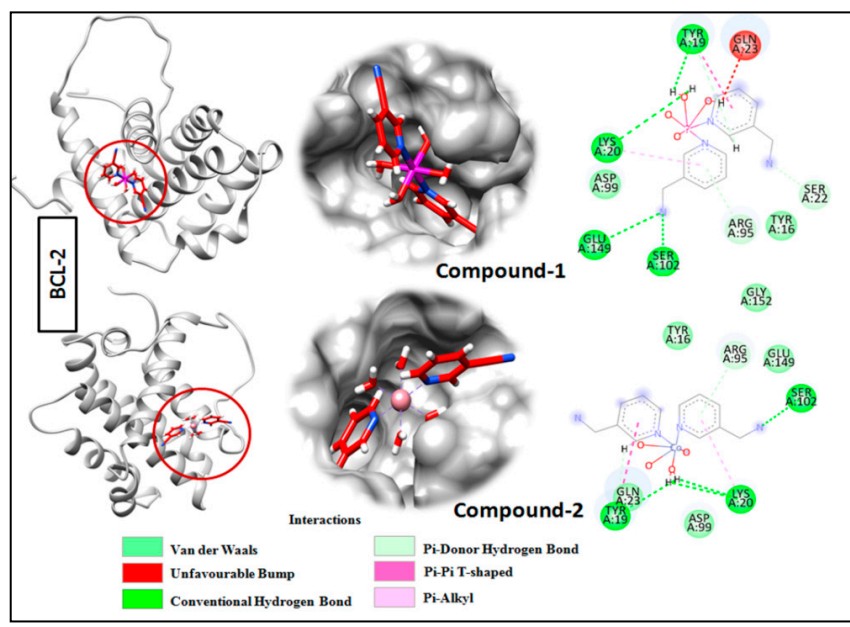

**Figure 15.** Docking analysis results for compounds **1** and **2** with BCL-2 receptor.

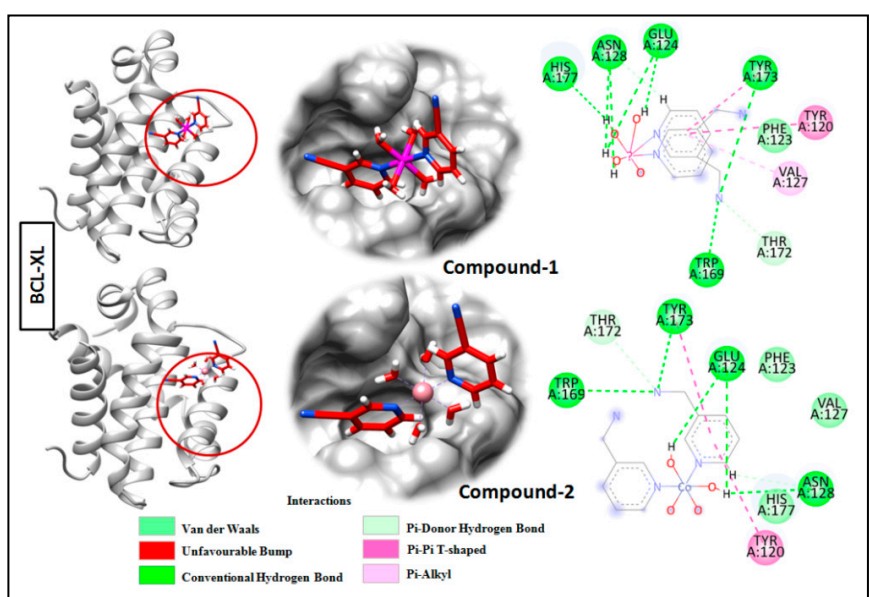

**Figure 16.** Docking analysis results of compounds **1** and **2** with BCL-XL receptor.

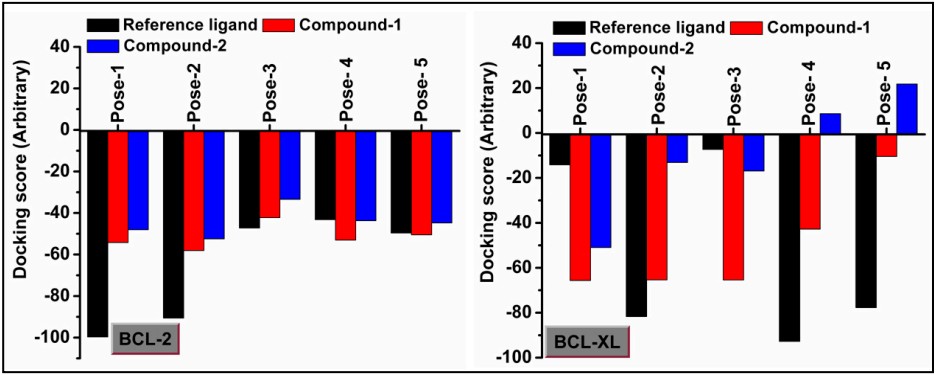

**Figure 17.** Docking scores of compounds **1** and **2** with BCL-2 and BCL-XL receptors.

## 3. Materials and Methods

The chemicals required for this study, viz. nickel(II) acetate tetrahydrate, cobalt(II) acetate tetrahydrate, 3-cyanopyridine, disodium salt of anthraquinone 1,5-disulfonic acid and sodium salt of 4-chlorobenzene sulfonic acid, were purchased from Sigma Aldrich and were used as received. C, H and N elemental analyses were conducted using a Perkin Elmer 2400 Series II CHN analyzer. FT-IR spectra of the compounds were recorded in the frequency range of 4000–500 $cm^{-1}$ in a Bruker Alpha II infrared spectrophotometer using KBr pellets. Using a Shimadzu UV spectrophotometer, the electronic spectra of the compounds were recorded. For the UV–Vis–NIR spectra, $BaSO_4$ powder was used. Room temperature magnetic moment values were determined at 300 K using a Sherwood Mark 1 Magnetic Susceptibility balance via the Evans method. In a Mettler Toledo TGA/DSC1 STAR$^e$ system, thermogravimetric analyses of the compounds were carried out under the flow of $N_2$ gas with a heating rate of 10 °C $min^{-1}$.

### 3.1. Synthesis

#### 3.1.1. Synthesis of [Ni(3-CNpy)$_2$(H$_2$O)$_4$]ADS·2.75H$_2$O (**1**)

Firstly, $Ni(CH_3COO)_2 \cdot 4H_2O$ (0.248 g, 1 mmol) and 3-CNpy (0.208 g, 2 mmol) were mixed in 10 mL of de-ionized water and stirred mechanically at room temperature for two hours. Then, 0.412 g of $Na_2$-ADS (1 mmol) was added to the reaction mixture with continuous stirring for another hour (Scheme 1). Then, the resulting solution was kept unperturbed for slow evaporation in cooling conditions (2–4 °C) for crystallization. After a few weeks, blue block shaped single crystals suitable for X-ray crystallographic analyses were obtained. The yield was 0.641 (84%). The anal. Calcd. For $C_{26}H_{27.5}N_4NiO_{14.75}S_2$ was as follows: C, 41.12%; H, 3.72%; N, 7.38%. The found percentages were C, 41.02%; H, 3.60%; N, 7.32%. The IR spectral data (KBr disc, $cm^{-1}$) were as follows: 3426(br), 2240(s), 1692(s), 1603(s), 1577(s), 1481(m), 1418(s), 1315(s), 1271(s), 1207(s), 1042(s), 971(s), 824(s), 729(s), 614(s) (s, strong; m, medium; w, weak; br, broad; sh, shoulder).

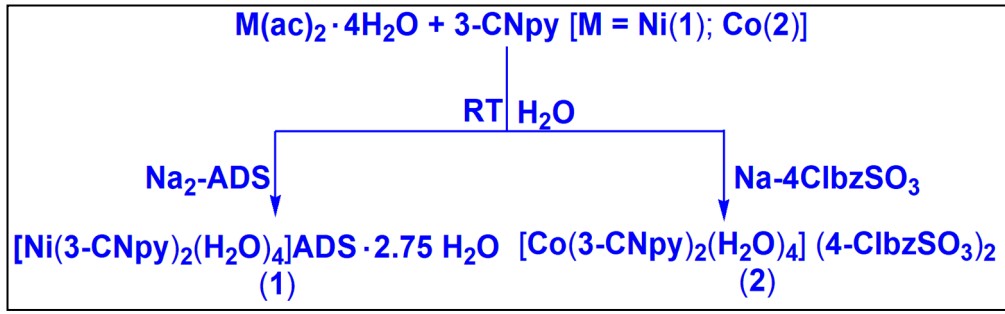

**Scheme 1.** Syntheses of compounds **1** and **2**.

#### 3.1.2. Synthesis of [Co(3-CNpy)$_2$(H$_2$O)$_4$] (4-ClbzSO$_3$)$_2$ (**2**)

The mixture of $Co(CH_3COO)_2 \cdot 4H_2O$ (0.249 g, 1 mmol) and 3-CNpy (0.192 g, 2 mmol) was dissolved in 10 mL of de-ionized water and stirred mechanically at room temperature for two hours. After two hours, Na-4-ClbzSO$_3$ (0.428 g, 2 mmol) was added to the resulting solution with stirring for another hour (Scheme 1). Then, the solution was left to evaporate slowly in a refrigerator at a temperature below 4 °C for crystallization. Red block shaped crystals suitable for single crystal X-ray diffraction were obtained by the slow evaporation of the mother liquor after several days. The yield was 0.609 g (84%). The anal. calcd. for $C_{24}H_{24}N_4O_{10}S_2Cl_2Co$ was as follows: C, 39.90%; H, 3.35%; N, 7.76%; the found percentages were C, 39.85%; H, 3.25%; N, 7.69%. The IR spectral data (KBr disc, $cm^{-1}$) were as follows: 3400(br), 3260(sh), 2233(s), 1909(m), 1685(s), 1609(s), 1488(s), 1418(s), 1195(s), 1124(s), 1092(s), 1035(s), 1003(s), 927(s), 824(s), 761(m), 646(s) (s, strong; m, medium; w, weak; br, broad; sh, shoulder).

### 3.2. Crystallographic Data Collection and Refinement

Suitable single crystals of compounds **1** and **2** were selected, covered with Parabar 10320 (formally known as Paratone N) and mounted on a Bruker D8 Venture diffractometer (Karlsruhe, Germany) with a Photon III 14 detector, using an Incoatec high brilliance IµS DIAMOND Cu tube equipped with Incoatec Helios MX multilayer optics. To minimize data and refine the cells, the Bruker APEX3 program was used. SADABS was used to scale and combine the various data sets for their wavelengths, as well as to perform semi-empirical absorption correction [105]. Using the WinGX [106] platform for personal computers, the crystal structures were solved directly, and then improved using the complete matrix least squares methodology with SHELXL-2018/3 [107]. Anisotropic thermal parameters were used to refine all non-hydrogen atoms using full-matrix least squares computations on $F^2$. Since hydrogen atoms have a low scattering power, it is challenging to find them precisely using X-ray data; as a result, hydrogen atoms were placed at predetermined sites and refined as riders. [108]. Non-coordinated water molecules in compound **1** were treated as disordered; OW5 presented a partial occupancy of 0.75, while OW6-OW7 and OW8-OW9 were found disordered over two complementary positions. The PLATON program was utilized to verify if the constructions exhibited better symmetry [109]. The Mercury program was used to prepare the graphics assets. [110]. The structural diagrams were drawn with Diamond 3.2 [111]. The crystallographic results for the compounds are tabulated in Table 2 and CCDC deposition numbers have been cited in Appendix A.

**Table 2.** Crystallographic data and structure refinement details for compounds **1** and **2**.

| Parameters | 1 | 2 |
|---|---|---|
| Formula | $C_{26}H_{27.5}N_4NiO_{14.75}S_2$ | $C_{24}H_{24}N_4O_{10}S_2Cl_2Co$ |
| Formula weight | 754.85 | 722.42 |
| Temp, [K] | 100.00 | 294.15 |
| Crystal system | Monoclinic | Monoclinic |
| Space group | $P2_1/c$ | $C2/m$ |
| a, [Å] | 14.3963(1) | 31.6924(9) |
| b, [Å] | 14.1160(1) | 6.9038(2) |
| c, [Å] | 14.8840(1) | 6.8614(2) |
| α, [°] | 90 | 90 |
| β, [°] | 98.551(2) | 95.4620(1) |
| γ, [°] | 90 | 90 |
| V, [Å$^3$] | 2991.1(4) | 1494.44(7) |
| Z | 4 | 2 |
| Absorption coefficient (mm$^{-1}$) | 2.973 | 7.993 |
| F(0 0 0) | 1558 | 738 |
| D (calcd), [Mg/m$^3$] | 1.676 | 1.605 |
| Index ranges | $-17 \leq h \leq 17, -16 \leq k \leq 17, -17 \leq l \leq 17$ | $-38 \leq h \leq 38, -8 \leq k \leq 8, -8 \leq l \leq 8$ |
| Crystal size, [mm$^3$] | $0.15 \times 0.09 \times 0.05$ | $0.18 \times 0.12 \times 0.06$ |
| θ range, [°] | 4.34 to 68.521 | 2.801 to 68.189 |
| Independent Reflections | 5408 | 1494 |
| Reflections collected | 30,922 | 9510 |
| Refinement method | Full-matrix leastsquares on $F^2$ | Full-matrix leastsquares on $F^2$ |
| Data/restraints/parameters | 5408/0/473 | 1494/0/128 |
| Goodness-of-fit on $F^2$ | 1.130 | 1.085 |
| Final R indices [I >2σ(I)] | R1 = 0.0445, wR2 = 0.1074 | R1 = 0.0311, wR2 = 0.0822 |
| R indices (all data) | R1 = 0.0473, wR2 = 0.1088 | R1 = 0.0313, wR2 = 0.0827 |
| Largest hole and peak [e·Å$^{-3}$] | −0.44 and 0.42 | −0.44 and 0.41 |

### 3.3. Computational Methods

The interaction energies of the compounds and supramolecular assemblies studied in this research were computed at the RI-BP86-D3/def2-TZVP [112,113] level of theory, using the experimental geometries and the program Turbomole 7.2 [114]. For both the Ni(II) and Co(II) octahedral environments, the high spin electronic configurations were used (two

and three unpaired electrons, respectively). No spin contamination was found in any of the systems. Grimme's D3 dispersion [112] correction was used, since it is convenient to correctly evaluate the non-covalent interactions investigated herein, and particularly those involving $\pi$-systems. The Bader's QTAIM method and NCI plot [115] reduced density gradient (RGD) isosurfaces were combined in the same representation to indicate the supramolecular interactions. RDG isosurfaces are useful to reveal non-covalent interactions in real space. The cubes needed to construct the NCI plot surfaces were generated at the same level of theory from the wavefunctions obtained using the Turbomole 7.2 program. The NCI Plot index isosurfaces correspond to both favorable and unfavorable interactions, as differentiated by the sign of the second density Hessian eigenvalue ($\lambda_2$) and defined by the isosurface color. In this study, we used red and yellow for repulsive interactions, and green and blue for attractive interactions. The QTAIM analysis [116] was carried out at the same level of theory by means of the MULTIWFN program [117] and represented by VMD software version 1.9 [118].

### 3.4. Cell Line and Drug Preparation

In vitro anticancer evaluation of the compounds **1** and **2** was carried out with the Dalton's lymphoma (DL) malignant cancer cell line. Dalton's lymphoma (DL) serves as an invaluable model in cancer research, due to its efficacy in preclinical studies for assessing both novel and established drugs in treating a range of cancers. Dalton's lymphoma is a transplantable T-cell lymphoma isolated by Dr. Dalton from the thymus of a dba 212 mouse at the National Cancer Institute [119]. Dalton's lymphoma stands as a rigorously characterized and reproducible transplantable tumor model, and is integral to drug development endeavors. Upon the introduction of DL cells into the abdominal cavity of healthy recipient mice, tumorigenesis promptly ensues with notable vigor. Maintenance of Dalton's lymphoma in laboratory settings is facilitated through ascites by serial transplantation in mice via intraperitoneal injection of $5 \times 10^5$ cells per mouse [120]. The DL cells were grown in a $CO_2$ incubator (Eppendorf, New brunswick Galaxy 170s, Hamburg, Germany) at 37 °C with 5% $CO_2$. The RPMI-1640 medium was supplemented with 10% fetal bovine serum (FBS), gentamycin (20 mg/mL), streptomycin (100 mg/mL) and penicillin (100 IU). In this study, 80% confluents of rapidly developing cells were sub-cultured. The different doses (0.01, 0.1, 0.5, 1, 5 and 10 μM) of the compounds were prepared in phosphate buffered saline (PBS; pH = 7.4). To observe the compounds' potential in normal healthy cells, peripheral blood mononuclear cells (PBMC) were used.

### 3.5. Cytotoxicity Study Using Trypan Blue Exclusion Method

The trypan blue exclusion assay is a well-known technique that is used to evaluate the cytotoxic behavior of compounds by selectively staining dead cells [121]. This assay is based on the fact that polar dyes like trypan blue are impermeable to viable cells with intact cell membranes, but can easily enter through damaged cell membranes of dead cells, thereby staining them blue. In the present study, the cytotoxic potential of eachcompound against DL cancer cells was determined using different concentrations (0.01, 0.1, 0.5, 1, 5 and 10 μM) using 96 cell culture plates (Thermo Scientific, Waltham, MA, USA, Cat. No: 265301). Following the staining process, viable and non-viable cells were identified based on their different staining colors. Based on the percentage cytotoxicity results, the $IC_{50}$ values, the half maximum inhibitory concentrations of the compounds were determined using the dose response non-linear curve fit model, following the equation form shown below:

$$y = A1 + (A2 - A1)/(1 + 10^{((LOGx0 - x) \times p)})$$

### 3.6. Molecular Docking Simulation

To support the findings of the wet lab based cytotoxicity assay, in silico molecular docking studies of the compounds were performed using fully functioning molegro virtual docker (Trial MVD 2010.4.0) software with anti-apoptotic target proteins [BCL-2 (PDB:

2O22) and BCL-XL (PDB: 2YXJ)] [122]. With a radius of 15 Å and a resolution of 0.35, the docking parameters were completed with 10 runs. The maximum interactions were 1500; the maximum population size was 50; the maximum steps were 300; the neighbor distance factor was 1.00; and the maximum number of poses returned were 5 [123]. To visualize the optimum docking orientations and their interactions, Chimera (https://www.cgl.ucsf.edu/chimera/ accessed on 8 April 2024) and Discovery Studio Visualization BIOVIA (https://www.3dsbiovia.com/products/collaborative-science/biovia-discovery-studio/, accessed on 8 April 2024) software programs were used [124].

*3.7. Statistical Analysis*

The experimental results are expressed as mean $\pm$ S.D., *n* = 3 (all measurements were repeated three times). Using an analysis of variance (ANOVA) (* $p \leq 0.05$), the data were analyzed, followed by the post hoc Tukey's range test.

**4. Conclusions**

Two new metal–organic multi-component Ni(II) and Co(II)compounds involving 3-cyanopyridine were synthesized and characterized using single crystal XRD, TGA, spectroscopic (IR, electronic) and elemental analyses. Further analysis of compound **1** unfolded the presence of antiparallel nitrile···nitrile and π-stacked assemblies involving alternate coordinated 3-CNpy and uncoordinated ADS and 4-ClbzSO$_3$ moieties. The O-atoms of the ADS moieties in **1** are involved in two different types of anion–π contacts involving the π-electrons of the aromatic ring and nitrile groups of the 3-CNpy moieties. Moreover, charge reverse Cl···N halogen bonding interactions were also observed in compound **2**. The structure guiding Cl···N halogen-bonded synthon is interesting, because the electron acceptor chlorine (halogen) atom, which is a σ-hole donor, is located in the anionic fragment, and the electron donor nitrogen of the nitrile moiety is located in the complex cationic fragment. We used MEP surfaces, combined NCI plots and QTAIM computational tools to analyze the supramolecular contacts in the solid-state assemblies of the compounds. Our studies revealed that the 'charge reverse' halogen-bonded assemblies in **2** are dominated by electrostatic Coulombic effects. In vitro antiproliferative evaluation of the compounds revealed that compounds **1** and **2** induce higher cytotoxicity in DL cells, with minimal effect on normal healthy PBMC cells. Molecular docking simulations of the compounds show strong binding affinities of the compounds with anti-apoptotic target proteins that are considerably involved in cancer cell growth.

**Supplementary Materials:** The following supporting information can be downloaded at: https://www.mdpi.com/article/10.3390/inorganics12040111/s1, Table S1: Selected bond lengths (Å) and bond angles (°) of Ni(II) and Co(II) centers in compounds **1** and **2**, respectively; Figure S1: Formation of supramolecular 1D chain of compound **1** assisted by anti-parallel CN···CN and C–H···N hydrogen bonding interactions; Figure S2: FT-IR spectra of compounds **1** and **2**; Figure S3: (a) UV–Vis–NIR spectrum of **1**, (b) UV–Vis spectrum of **1**; Figure S4: (a) UV–Vis–NIR spectrum of **2**, (b) UV–Vis spectrum of **2**; Figure S5: Thermogravimetric curves of compounds **1** and **2**.

**Author Contributions:** Conceptualization, A.F., A.K.V. and M.K.B.; methodology, A.F., A.K.V. and M.K.B.; software, A.F., R.M.G. and A.K.V.; formal analysis, A.F.; investigation, S.B., T.B., J.D. and R.M.G.; data curation, M.B.-O.; writing—original draft preparation, S.B.; writing—review and editing, M.K.B.; visualization, A.F.; supervision, M.K.B.; project administration, A.F. and M.K.B.; funding acquisition, A.F. and M.K.B. All authors have read and agreed to the published version of the manuscript.

**Funding:** Financial support was provided by SERB-SURE (grant number: SUR/2022/001262); ASTEC, DST, Govt. of Assam (grant number ASTEC/S&T/192(177)/2020-2021/43); the Gobierno de Espana, MICIU/AEI (project number PID2020-115637GB-I00); and the Department of Biotechnology (DBT), Government of India (project no. BT/INF/22/SP45376/2022), all of whom are gratefully acknowledged. S.B thanks CSIR [09/1236(17224)/2023-EMR-I], Govt. of India for Junior Research Fellowship (JRF). The authors thank IIT-Guwahati for the TG data.

**Data Availability Statement:** Data are contained within the article and Supplementary Materials.

**Conflicts of Interest:** The authors declare no conflicts of interest. The funders had no role in the design of the study; in the collection, analyses or interpretation of data; in the writing of the manuscript; or in the decision to publish the results.

**Appendix A**

CCDC 2163547 and 2163548 contains the supplementary crystallographic data for the compounds **1** and **2**. These data can be obtained free of charge at http://www.ccdc.cam.ac.uk or from the Cambridge Crystallographic Data Centre, 12 Union Road, Cambridge CB2 1EZ, UK; fax: (+44) 1223-336-033; or E-mail: deposit@ccdc.cam.ac.uk.

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
