# Peer review of "‘Charge Reverse’ Halogen Bonding Contacts in Metal-Organic Multi-Component Compounds: Antiproliferative Evaluation and Theoretical Studies"

_inorganics, doi:10.3390/inorganics12040111_

Round 1

Reviewer 1 Report

Comments and Suggestions for Authors

Authors report on the synthesis, characterization, detailed XRD structure analysis, as well as bioevaluation and molecular docking studies on metal complexes with Ni and Co as central metals. Authors are reporting on an interesting “charge reverse” halogen bonding in these structures.

My main concern is regarding interpretation of non-covalent interactions. There seems to be some discrepancies between data in the manuscript with data obtained in cif files. Authors should check that. Also, it seems authors didn’t use Platon as a tool for analysis of non-covalent interactions and it seems they used some arbitrary values in their interpretation. This is very problematic point. Authors should recheck all claims regarding non-covalent interactions and support their finding with Platon program.

 Additional comments:

Ÿ - In Table 3 in many cases standard deviations are missing. Check Platon analysis.

Ÿ  - There seems to be some discrepancies between data in Table 3 and Platon analysis of cif files deposited to CCDC. For example, compound 1 - for C3-H3…N4 data in Table 3 is 3.402, however, Platon analysis reports 3.411(4). Also, in Table 3 is reported C8-H8…N3 interaction, however, Platon analysis doesn’t report this interaction as H-bonding. Also, authors should check the C15-H15…O14 interaction since it has very small angle and is also not reported by Platon analysis. Authors should double check all H-bonds reported in Table 3 and follow the Platon analysis.

Ÿ  - Check also Table 3 for compound 2 – Platon analysis reports only three H-bonds, two with O1W and one with C6A (this one with very small angle). Authors should double check all H-bonds reported in Table 3 and follow the Platon analysis.

Ÿ  - Formula of compound 2 is not given correctly. Since this is a ionic compound it should be given as Co(3-CNpy)2(H2O)4](4-ClbzSO3)2 (as we report calcium chloride as CaCl2 and not Ca 2Cl.

-Ÿ  For the refinement of structures, a SADI commend was used and also some parts are treated as disordered. Info about this is missing in the section 2.3.

Author Response

First, we would like to thank this reviewer for his/her careful reading of the manuscript, corrections and suggestions. Our "point-by-point" relies follow:

Comments and Suggestions for Authors

Authors report on the synthesis, characterization, detailed XRD structure analysis, as well as bio-evaluation and molecular docking studies on metal complexes with Ni and Co as central metals. Authors are reporting on an interesting “charge reverse” halogen bonding in these structures.

 Q1. My main concern is regarding interpretation of non-covalent interactions. There seems to be some discrepancies between data in the manuscript with data obtained in cif files. Authors should check that. Also, it seems authors didn’t use Platon as a tool for analysis of non-covalent interactions and it seems they used some arbitrary values in their interpretation. This is very problematic point. Authors should recheck all claims regarding non-covalent interactions and support their finding with Platon program.

Reply: We have checked the non-covalent interactions with Platon program and make necessary modifications. However, a few additional hydrogen bonding interactions are also observed in the Diamond 3.2 software at acceptable distances which has been used to finalize the packing diagrams. 

 Additional comments:

Q2. - In Table 3 in many cases standard deviations are missing. Check Platon analysis.

Reply: Necessary modifications have been done in the revised manuscript as suggested by the esteemed reviewer.

Q3.  - There seems to be some discrepancies between data in Table 3 and Platon analysis of cif files deposited to CCDC. For example, compound 1 - for C3-H3…N4 data in Table 3 is 3.402, however, Platon analysis reports 3.411(4). Also, in Table 3 is reported C8-H8…N3 interaction, however, Platon analysis doesn’t report this interaction as H-bonding. Also, authors should check the C15-H15…O14 interaction since it has very small angle and is also not reported by Platon analysis. Authors should double check all H-bonds reported in Table 3 and follow the Platon analysis.

Reply: Table 3 has been modified in the revised manuscript. The hydrogen bonding interactions with small angle mentioned above have been removed.

Q4.  - Check also Table 3 for compound 2 – Platon analysis reports only three H-bonds, two with O1W and one with C6A (this one with very small angle). Authors should double check all H-bonds reported in Table 3 and follow the Platon analysis.

Reply: As mentioned in aforementioned query, we have added a few additional hydrogen bonds apart from those reported in the Platon program. These additional hydrogen bonding interactions are observed in the diamond software which provides additional reinforcement to the crystal structure.

Q5. - Formula of compound 2 is not given correctly. Since this is a ionic compound it should be given as Co(3-CNpy)2(H2O)4](4-ClbzSO3)2 (as we report calcium chloride as CaCl2 and not Ca 2Cl.

Reply: Formula of compound 2 has been revised accordingly.

Q6. - For the refinement of structures, a SADI commend was used and also some parts are treated as disordered. Info about this is missing in the section 2.3.

Reply: SADI command has been deleted in the revised crystal structures. The information regarding the disordered water molecules have been incorporated in the revised manuscript.

Reviewer 2 Report

Comments and Suggestions for Authors

Please correct some information in Table 1. The Formula for compound 1, 2 is wrong, and also formula weight.

What is the wavelength for measurement?

Please do absorption correction for two data sets and refine once again.

After correction , change in cif files and upload once again cif files.

Table 1.

Formula

C104H110N16O59S8Ni4

C24H24N4O10S2Cl2Co

Formula weight

3019.39

722.42

Table 2. with Selected bond lengths (A) and bond angles (°) of Ni(II) and Co(II) centers in the compounds is absolutely unnecessary, especially authors claim that geometry is similar with previously reported complexes. (line 248/249). Please remove it to supplementary materials.

How the” charge reverse” halogen bond could affect on some futures in the crystal lattice or for cytotoxicity?

Author Response

First, we would like to thank this reviewer for his/her careful reading of the manuscript, corrections and suggestions. The point-by-point responses follow:

Comments and Suggestions for Authors

 Q1. Please correct some information in Table 1. The Formula for compound 1, 2 is wrong, and also formula weight.

Reply: Table 1 has been revised according to the revised cifs of the compounds

Q2. What is the wavelength for measurement?

Reply: The wavelength for the measurement of the crystal structures has been included in the revised manuscript.

Q3. Please do absorption correction for two data sets and refine once again.

Reply: Absorption corrections have been performed and the structures are refined again.

Q4. After correction, change in cif files and upload once again cif files.

Reply: We have uploaded the new cifs during revision submission; also we have resubmitted the structures in CCDC

Q5. Table 1.

Formula

C104H110N16O59S8Ni4, C24H24N4O10S2Cl2Co

Formula weight

3019.39, 722.42

Reply: The Formula and formula weight of the compounds have been included in Table 1 as per the revised cifs.

Q6. Table 2. with Selected bond lengths (A) and bond angles (°) of Ni(II) and Co(II) centers in the compounds is absolutely unnecessary, especially authors claim that geometry is similar with previously reported complexes. (line 248/249). Please remove it to supplementary materials.

Reply: Table 2 has been shifted to supplementary materials.

Q7. How the” charge reverse” halogen bond could affect on some futures in the crystal lattice or for cytotoxicity?

Reply: As mentioned in the introduction of the manuscript; halogen bonding interactions, a subset of σ-hole interaction, possess a very important role in the stabilization of the supramolecular assemblies [Chem. Rev. 2016,116, 2478–2601; Politzer, P.; Lane, P.; Concha, M.C.; Ma, Y.; Murray, J.S. J. Mol. Model.2007,13, 305–311; Riley, K.E.; Murray, J.S.; Fanfrlík, J.; Řezáč, J.; Solá, R.J.; Concha, M.C.; Ramos, F.M.; Politzer, P. J. Mol. Model. 2011, 17, 3309–3318]. However; the reports of “charge reverse” halogen bonding; where electron acceptor halogen atom is located on the electron rich fragment (anion) and the electron donor atom on the electron poor fragment (cation); is still scarce in the literature. The formation of such unusual charge reversed halogen bonding may unfold new avenues in the crystal engineering of molecular solids in near future.

As the concept of “charge reverse” halogen bond is pretty new in the field of supramolecular chemistry, hence it is a bit difficult to predict its role in cytotoxic activities. However, as we all know, DNA or RNA of cells are negative in nature, hence there might be a probability of formation of non-covalent charge reverse halogen bonding involving the amino N-atom of DNA of cancerous cells. In such a case, DNA damage would takes place which will ultimately results in cancer cell death.

Round 2

Reviewer 1 Report

Comments and Suggestions for Authors

Corrections are adequate.

Author Response

Thank you for your recommendation

Reviewer 2 Report

Comments and Suggestions for Authors

Were the changes done as the authors claim in the author's reply form?

I do not see it in Table 1

"C104H110N16Ni4O59S8" 

"Formula weight 3019.39"

I do not recommend accepting unless the authors correct it.

Author Response

We thank the referee for his/her second reading of the manuscript and further corrections. We are sorry for the omission. Our reply follows:

Comments and Suggestions for Authors

Q1. Were the changes done as the authors claim in the author's reply form?

I do not see it in Table 1

"C104H110N16Ni4O59S8" 

"Formula weight 3019.39"

I do not recommend accepting unless the authors correct it.

Reply: Yes, changes were done as pointed out by the esteemed reviewers except the molecular formula, formula mass and the corresponding Z value of compound 1 which got overlooked and was not changed. We have now corrected and incorporated the changes in the revised manuscript. We thank esteemed reviewer for pointing out the mistake.